# OpenReview forum: "Concept Removal Guidance: Evidence-Calibrated Negative Guidance for Safe Diffusion Sampling"
_ICML.cc/2026/Conference — ICML 2026 spotlight_

### Official Review · Reviewer_RUtA · 2026-03-02

**Soundness:** 3
**Presentation:** 3
**Significance:** 3
**Originality:** 3
**Overall Recommendation:** 5
**Confidence:** 2

**Summary:**

The paper introduces Concept Removal Guidance (CRG), a training-free, inference-time method for suppressing unwanted concepts in text-to-image diffusion models. Unlike standard fixed negative prompting, which creates a safety–fidelity trade-off, and prior dynamic or classifier-based methods, CRG estimates the presence of an undesirable concept at each diffusion step directly from the model’s own noise predictions and predicted clean image. It then adaptively scales negative guidance using a constrained update that enforces a target suppression level while minimally altering the intended generation. Experiments show that CRG improves robustness to adversarial prompts and preserves image quality without requiring fine-tuning or external classifiers.

**Compliance With Llm Reviewing Policy:**

Affirmed.

**Final Justification:**

The authors addressed all my concers. Raising my score to accept.

**Key Questions For Authors:**

See weaknesses

**Limitations:**

The authors did not provide a limitations section in the current manuscript.

**Strengths And Weaknesses:**

Strengths

* The paper is clearly written, well-structured.
* The proposed method is novel and technically well grounded, with solid mathematical support.
* CRG is training-free and plug-and-play, which makes it practical and easy to integrate into existing diffusion pipelines without additional models or fine-tuning.
* The empirical evaluation is comprehensive, showing strong performance across multiple benchmarks.

Weaknesses
* The approach still depends on manually crafted negative prompts, which can be nontrivial to design in practice. Are there automated or adaptive strategies for constructing them?
* The evaluation relies solely on automated metrics, which are imperfect proxies for human judgment in safety and image quality. A user study would better substantiate the claimed improvement in safety–fidelity trade-off.
* Additional analysis of computational overhead and generalization to broader real-world prompt distributions would further strengthen the work.

---

> ### Author Rebuttal · Authors · 2026-03-31
>
> ## Automated or Adaptive Strategies for Constructing Negative Prompts
> Designing negative prompts is inherently challenging because there are often many candidate terms related to a given target concept, and it is not straightforward to determine which of them will effectively suppress that concept. However, our framework already provides a principled and adaptive way to address this issue through the proposed Concept Presence (CP) metric. As described in Appendix H, we first construct a candidate pool from prior safety literature, evaluate the CP of each candidate term on a subset of adversarial prompts, and then select the final negative prompt based on the strongest CP signals.
> Importantly, this is not only a heuristic description: when the resulting CP-selected prompts are applied to baselines such as SAFREE and TraSCE, they yield better removal performance than the original prompts used in those methods as shown in **Table 6**. Thus, while the final prompt set in the current submission is manually specified, the proposed framework already supports a practical automated and adaptive negative-prompt construction pipeline.
>
> **Table 6. Comparison on P4D, UnlearnAtk, MMA, and COCO metrics. Lower is better for P4D, UnlearnAtk, MMA, and FID, while higher is better for CLIP.**
> | Method | P4D ↓ | UnlearnAtk ↓ | MMA ↓ | COCO CLIP ↑ | COCO FID ↓ |
> |---|---:|---:|---:|---:|---:|
> | SAFREE | 0.529 | 0.275 | 0.611 | 31.11 | 55.79 |
> | SAFREE + CRG Prompt | 0.423 | 0.232 | 0.570 | 30.94 | 56.59 |
> | TraSCE | 0.291 | 0.134 | 0.490 | 30.67 | 57.11 |
> | TraSCE + CRG Prompt | 0.278 | 0.113 | 0.431 | 30.69 | 58.10 |
>
> More broadly, we also note that recent work [2] has explored the use of vision-language models for adaptive prompt refinement. We view such approaches as complementary to our method, and integrating VLM-based prompt generation with CP-based term selection is a promising direction for future work.
>
> [2] VLM-Guided Adaptive Negative Prompting for Creative Generation
>
> ---
> ## Automated Evaluation Metrics
> The reliance on automated metrics is an intentional design choice for fair comparison, not a weakness of the paper. In prior concept removal works like TraSCE and SAFREE, evaluation is typically conducted using classifier-based metrics that are tailored to the task, reporting accuracy ($\mathrm{Acc}_e$ and $\mathrm{Acc}_u$) for artist style removal, and attack success rate for NSFW concept removal tasks such as nudity and violence. We follow the same evaluation protocol so that CRG is assessed under the same criteria used for existing baselines, which is essential for a meaningful empirical comparison.
> Accordingly, we view classifier-based evaluation as the correct evaluation standard in this setting. To further reinforce the evidence, we also include additional artist style removal and identity removal evaluations using multiple classifier models beyond GPT-4o. As reported in **Table 2** and **Table 4** in [[link]](https://anonymous.4open.science/r/Rebuttal-CRG-B0EF/README.md), the conclusions remain consistent across classifier models, which further strengthens the empirical support for our results.
>
> ---
> ## Analysis of Computational Overhead
> As shown in **Table 7** in [[link]](https://anonymous.4open.science/r/Rebuttal-CRG-B0EF/README.md), both DNG and CRG incur additional computational cost because they require one extra diffusion-model noise prediction relative to the backbone model and negative prompting. Nevertheless, they are still more efficient than gradient-based approaches such as TraSCE.
>
> ---
> ## Generalization to broader real-world prompt distributions
> We further report experimental results on PartiPrompts in **Table 8** in [[link]](https://anonymous.4open.science/r/Rebuttal-CRG-B0EF/README.md), a comprehensive benchmark consisting of 1,632 English prompts that span diverse categories and varying levels of complexity, from simple descriptions to compositionally challenging queries. The effect of nudity removal observed on the PartiPrompts benchmark is consistent with the trends seen on COCO-1k prompt dataset.
>
>
> ---
> ## Limitations
> While CRG demonstrates strong empirical performance, we acknowledge several limitations. First, as an inference-time mitigation, CRG serves as a complementary safety layer; it may remain vulnerable to highly implicit or adaptive adversarial prompts. In addition, the effectiveness of CRG depends on user-specified negative prompts, which can be challenging to design in open-vocabulary settings. Extending CRG with automated or adaptive negative-prompt construction is left for future work. Finally, our proposed concept-presence score utilizes modeling approximations and should be interpreted as a surrogate signal rather than a definitive measure of semantic harm.

---

> > ### Author Rebuttal · Reviewer_RUtA · 2026-04-03
> >
> > Thank you for the detailed rebuttal. Some of my concerns have been addressed. After reading the rebuttal and the other reviewers’ comments, I am maintaining my score. First, I find the claim of “automated or adaptive” prompt construction to be overstated. Their pipeline is not fully automatic in the strong sense, as it still depends on a manually curated candidate pool drawn from prior safety literature. In addition, human evaluation is still not adequately addressed. Safety and fidelity are only imperfectly captured by classifiers. Therefore, although prior work did not include a user study, I still find such a study useful and believe it would strengthen the paper.

---

> > > ### Author Response · Authors · 2026-04-08
> > >
> > > Thank you for your careful and constructive feedback.
> > >
> > > ## Regarding Negative Prompts
> > >
> > > In the original version of our paper, we selected negative prompts from prior literature. To address this concern more directly, we added experiments using negative prompts automatically generated by LLM.
> > > Specifically, we asked Grok to generate eight keywords most closely related to nudity and sexuality, and directly used them as negative prompts. Even with purely LLM-generated negative prompts, CRG remains effective and continues to perform strongly relative to the compared baselines.
> > >
> > > | Negative Prompts | Algorithm | P4D ↓ | Unlearn ↓ | MMA-Diff ↓ | CLIP ↑ | FID ↓ |
> > > |---|---|---:|---:|---:|---:|---:|
> > > | - | SD v1.4 | 0.854 | 0.655 | 0.949 | 31.10 | - |
> > > | Handcrafted | SAFREE | 0.529 | 0.275 | 0.611 | **31.11** | 55.79 |
> > > | Handcrafted | TraSCE | 0.291 | *0.134* | 0.490 | 30.67 | 57.11 |
> > > | Handcrafted | NP | 0.338 | 0.183 | 0.453 | 30.49 | *51.74* |
> > > | Handcrafted | DNG | *0.225* | 0.183 | **0.139** | 30.79 | 55.97 |
> > > | Handcrafted | CRG (OURS) | **0.026** | **0.059** | *0.164* | *30.85* | **36.05** |
> > > | GROK | SAFREE | 0.384 | 0.190 | 0.596 | *30.81* | 56.09 |
> > > | GROK | TraSCE | 0.265 | *0.120* | 0.508 | 30.60 | 57.91 |
> > > | GROK | NP | 0.331 | 0.176 | 0.423 | 30.35 | *51.79* |
> > > | GROK | DNG | *0.245* | 0.218 | **0.229** | 30.76 | 56.36 |
> > > | GROK | CRG (OURS) | **0.053** | **0.049** | *0.271* | **30.87** | **40.17** |
> > >
> > > [Grok 8 keywords] Nudity, Naked, Erotic, Sensual, Explicit, Sexual, Lewd, NSFW
> > >
> > > ## Regarding Human Evaluation
> > > We additionally conducted a human evaluation on the artist style removal task with 30 participants, each evaluating 100 images per method, using the same protocol as the GPT-4o-based assessment (Section 4.2). Participants were shown images generated by each method and asked to identify the artist style from five options. Van Gogh was the target style to be removed, while Pablo Picasso, Andy Warhol, Rembrandt, and Caravaggio were non-target styles. We report $Acc_e$, the accuracy on the erased style (lower is better), and $Acc_u$, the accuracy on the non-target styles (higher is better).
> > >
> > > **Human evaluation results for artist-style removal on Stable Diffusion v1.4.**
> > > || Remove|“VanGogh” |
> > > |---|---:|---:|
> > > | Method | $ACC_e$ ↓| $ACC_u$ ↑|
> > > | CFG | 0.837 | 0.803 |
> > > | NP | 0.376 | *0.816* |
> > > | SAFREE | 0.418 | 0.814 |
> > > | TraSCE | 0.434 | 0.815 |
> > > | DNG | *0.304* | 0.724 |
> > > | OURS | **0.259** | **0.820** |
> > >
> > > As shown in the above Table, our method yields the lowest erased style accuracy (0.259) together with the highest non-target accuracy (0.820). This human evaluation is aligned with the GPT-4o-based assessment, indicating that the same selectivity trend is also reflected in human judgments. These results provide additional evidence that our method effectively removes the target style without unnecessarily harming non-target styles.
> > > We note, however, that we were not able to conduct a human evaluation for the nudity removal setting, since such a study would require exposing participants to sensitive sexual content. This raises practical and ethical challenges for conducting the evaluation safely and appropriately. We therefore leave this as an important future work.

---

### Official Review · Reviewer_TE7Y · 2026-03-09

**Soundness:** 2
**Presentation:** 3
**Significance:** 3
**Originality:** 2
**Overall Recommendation:** 3
**Confidence:** 4

**Summary:**

This paper introduces Concept Removal Guidance (CRG), a training-free, plug-and-play inference-time defense for avoiding unwanted concept generation in text-to-image diffusion models. It first estimates concept presence at each denoising step using only the model's noise predictions, and then derives a closed-form update that calibrates negative guidance based on whether the estimated concept presence exceeds a threshold. Experimental results demonstrate its effectiveness in avoiding nudity and art styles in several diffusion models.

**Compliance With Llm Reviewing Policy:**

Affirmed.

**Key Questions For Authors:**

See the weaknesses.

**Limitations:**

Yes

**Strengths And Weaknesses:**

Strengths:
1. The method is training-free and plug-and-play, demonstrating effective generalization across diverse diffusion models.
2. The related work provides informative context, and the experiments validate its efficacy against multiple baselines.
3. The paper is well-structured and clearly written, making it easy to follow.

Weaknesses:

Major:
1. The core derivation of the concept-presence estimation equations lacks rigorous justification. Specifically, the paper does not adequately explain: (1) the transition from the likelihood-based definition (Eq. 5-6) to the noise-prediction formulation (Eq. 7-9), and (2) the connection between concept presence and the mutual information & KL divergence terms in Eqs. 5-6.
2. The experimental evaluation is inadequate. The evaluation is limited to only nudity and two artistic styles, lacking diversity in concept types. Besides, for art style removal assessment, the reliance on only one classifier (unspecified) is not convincing. Incorporating additional evaluation metrics, including human judgement, would strengthen the validation of the method's effectiveness.
3. While ablation studies are provided, the method's performance depends on multiple key hyperparameters like the threshold τ and the scaling factor λ₀. Finding optimal settings for new concepts or models might require non-trivial tuning, potentially limiting its plug-and-play utility in practice.
4. The method only evaluates single-target removal. It lacks compositional multi-concept removal evaluation to further demonstrate its effectiveness.

Minor:
1. For style removal evaluation, the classfier for computing classification accuracy is not specified in the paper.
2. In the abstract, it mentions suppressing targets such as “artist style and violence”. However, I think there is no violence-related experiment in the paper.
3. The paper lacks analysis of computational efficiency, specifically runtime comparisons with baselines.
4. The title in the PDF submission differs from the title shown on the OpenReview platform.

---

> ### Author Rebuttal · Authors · 2026-03-31
>
> ## Regarding Eq. (5) and Eq. (6)
> First, we defined the *concept presence* (CP) as
> $$CP(c- |c)\triangleq\mathbb{E}\_{p_\theta(x_0|c)}\left[\log p(c- |x_0)\right],$$
> which measures the expected log-posterior probability of the negative concept $c-$ being present in images generated under prompt $c$.
> Applying Bayes' rule,
> $$\log p(c-|x_0)=\log p(c-)+\log p(x_0|c-)-\log p(x_0).$$
> Since the true data distributions are not directly accessible, we approximate them using the corresponding model-induced distributions of the pretrained diffusion model, i.e.,
> $p(x_0|c-)\approx p_\theta(x_0|c-)$ and $p(x_0)\approx p_\theta(x_0)$.
> Substituting this into the definition gives
> $$CP(c-|c)\approx\log p(c-)+\mathbb{E}\_{p_\theta(x_0|c)}\left[\log\frac{p_\theta(x_0|c-)}{p_\theta(x_0)}\right].$$
> We then add and subtract $\log p_\theta(x_0|c)$ inside the log-ratio:
> $$\log\frac{p_\theta(x_0|c-)}{p_\theta(x_0)}=\log\frac{p_\theta(x_0|c)}{p_\theta(x_0)}-\log\frac{p_\theta(x_0|c)}{p_\theta(x_0|c-)}.$$
> Therefore,
> $$CP(c-|c)\approx{\mathbb{E}\_{p_\theta(x_0|c)}\left[\log \frac{p_\theta(x_0|c)}{p_\theta(x_0)}\right]}
> -{\mathbb{E}\_{p_\theta(x_0|c)}\left[\log\frac{p_\theta(x_0|c)}{p_\theta(x_0|c-)}\right]}+\log p(c-).$$
> Since $\log p(c-)$ is constant with respect to $c$, it does not affect our inference-time control rule.
>
> ## Regarding Eq. (6) to Eq. (8)
> For additional details, please see our response **Regarding Eq (6) to Eq (8)** to Reviewer “cgQt”, where we address this point more fully.
>
> ## Explanation about Experimental Evaluation Process
> For artist style removal, we use GPT-4o for evaluation, following prior works (SAFREE and TraSCE). For nudity removal, we employ NudeNet, regarding image as unsafe when its classification score on NudeNet exceeds a threshold of 0.6.
>
> ## Additional Artist Style Removal Assessment
> Our style-removal evaluation was intentionally conducted under the same automated evaluation setting used in prior work, rather than under a new and unmatched protocol. The paper already follows prior style-removal works and reports $Acc_e$ and $Acc_u$, which directly measure the two quantities that matter most here: successful removal of the target style and preservation of unrelated styles. We therefore view classifier-based evaluation as the correct baseline for fair comparison, not as a weakness specific to our paper. At the same time, to further strengthen the evidence, we report additional style-removal results in **Table 4** in [[link]](https://anonymous.4open.science/r/Rebuttal-CRG-B0EF/README.md) using multiple evaluator models beyond GPT-4o.
>
> ## Diverse Concept Removal
> For a discussion on Diverse Concept Removal, we refer the reviewer to our response to Reviewer “cgQt” regarding **Additional Concept Removal**.
>
> ## Hyperparameter tuning and usability
> The presence of hyperparameters should not be conflated with impracticality. CRG requires only lightweight inference-time tuning, which is fundamentally different from retraining-based methods and far cheaper in both computation and engineering cost. In fact, we already report extensive hyperparameter analysis for $k,\tau$, and $\lambda_0$ in Appendix E, showing both their interpretable roles and their empirical stability across meaningful ranges.
>
> ## Compositional Multi-concept Removal
> We further evaluate a compositional multi-concept removal setting, where CRG-Multi suppresses both violence and nudity via sequential projection using $CP_k(c_1- |\hat{x}_0)$ and $CP_k(c_2- |\hat{x}_0)$. As shown in **Table 5** in [[link]](https://anonymous.4open.science/r/Rebuttal-CRG-B0EF/README.md), this extension is effective on both violence and nudity benchmarks, demonstrating that CRG is not limited to single-target removal, while introducing only a modest computational overhead and a moderate quality trade-off.
>
> ## Analysis on Computational Efficiency
> As shown in **Table 7** in [[link]](https://anonymous.4open.science/r/Rebuttal-CRG-B0EF/README.md), both DNG and CRG incur additional computational cost because they require one extra diffusion-model noise prediction relative to the backbone model and negative prompting. Nevertheless, they are still more efficient than gradient-based approaches such as TraSCE.

---

> > ### Author Rebuttal · Reviewer_TE7Y · 2026-04-04
> >
> > Thank you for the rebuttal. After reviewing your responses, I still stand by my original decision, so my score remains unchanged.

---

> > > ### Author Response · Authors · 2026-04-08
> > >
> > > Thank you for the careful review and your acknowledgement of our response. We appreciate that our response helped clarify the main concerns. We will incorporate the feedback and our responses into the final paper.
> > > ## Regarding Human Evaluation
> > > We additionally conducted a human evaluation on the artist style removal task with 30 participants, each evaluating 100 images per method, using the same protocol as the GPT-4o-based assessment (Section 4.2). Participants were shown images generated by each method and asked to identify the artist style from five options. Van Gogh was the target style to be removed, while Pablo Picasso, Andy Warhol, Rembrandt, and Caravaggio were non-target styles. We report ($Acc_e$), the accuracy on the erased style (lower is better), and ($Acc_u$), the accuracy on the non-target styles (higher is better).
> > >
> > > **Human evaluation results for artist-style removal on Stable Diffusion v1.4.**
> > > || Remove|“VanGogh” |
> > > |---|---:|---:|
> > > | Method | $ACC_e$ ↓| $ACC_u$ ↑|
> > > | CFG | 0.837 | 0.803 |
> > > | NP | 0.376 | *0.816* |
> > > | SAFREE | 0.418 | 0.814 |
> > > | TraSCE | 0.434 | 0.815 |
> > > | DNG | *0.304* | 0.724 |
> > > | OURS | **0.259** | **0.820** |
> > >
> > > As shown in the above Table, our method yields the lowest erased style accuracy (0.259) together with the highest non-target accuracy (0.820). This human evaluation is aligned with the GPT-4o-based assessment, indicating that the same selectivity trend is also reflected in human judgments. These results provide additional evidence that our method effectively removes the target style without unnecessarily harming non-target styles.
> > > We note, however, that we were not able to conduct a human evaluation for the nudity removal setting, since such a study would require exposing participants to sensitive sexual content. This raises practical and ethical challenges for conducting the evaluation safely and appropriately. We therefore leave this as an important future work.

---

### Official Review · Reviewer_qxe7 · 2026-03-12

**Soundness:** 4
**Presentation:** 4
**Significance:** 3
**Originality:** 3
**Overall Recommendation:** 5
**Confidence:** 3

**Summary:**

This paper proposes a training-free guidance method, Concept Removal Guidance (CRG, to remove unsafe inference-time generation in text-to-image tasks. It first derives *concept presence (CP)* and then rewrites the classifier guidance terms as conditional probabilities over the negative concept using Bayes's theorem. Then, they estimate the CP using a look-ahead estimate of a clean image (Tweedie’s formula) and derive the dynamic guidance signal. Lastly, the authors demonstrate that the CRG can apply to nudity concept removal and artist style removal while maintaining high fidelity.

**Compliance With Llm Reviewing Policy:**

Affirmed.

**Final Justification:**

As mentioned in **Rebuttal Acknowledgment**. Thanks.

I appreciate the authors' clever use of the guidance technique to address the safety scenario, but the limitation of user-specified negative prompts is critical for this field.

**Key Questions For Authors:**

1. If I understand correctly, $$CP_k(c- \vert \hat{x}^\theta_0(x_t, c))$$ could also be applied to other methods, right? If so, we can compare the different concept removal methods based on CP, which quantifies the generating images affects by the original concept or distinguishable of the original concept from negative concept.

**Limitations:**

yes

**Strengths And Weaknesses:**

_Soundness_

1. The derivation of CP, estimation of CP, and guidance signal (Eq. (17) with dynamic concept removal scale $\omega_\mathrm{CRG}$) are clear and sensible.
2. The experimental results covers different inference-time concept removal methods and SOTA models (SDXL and SD v3).

_Presentation_

1. The storyline is clear and good. In particular, the authors precent CRG could robustly achieve robustly concept removal effects but keep high fidelity on qualitative and quantitive results.

_Significance_

1. With rising of fine-tuned text-to-image diffusion models released on the public space, the controllable generation for safety-criteria becomes important. The authors propose guidance technique for concept removal brings good angle to the field.

_Originality_

1. The authors carefully design CRG with existing theoretical works for diffusion models such as posterior sampling, compositional generation to present their work's benefits from existing works.

---

> ### Author Rebuttal · Authors · 2026-03-31
>
> ## Applying CP to other methods
> We agree that $CP_k(c-|\hat{x}\_0^\theta(x_t,c))$ is not specific to CRG and can more broadly serve as a general concept-presence signal for other inference-time concept removal methods. Motivated by this observation, we additionally evaluate a variant of TraSCE augmented with our CP-based signal. Specifically, we use the estimated concept presence to adaptively modulate the strength of TraSCE’s gradient-based guidance, assigning a larger guidance weight $\lambda_{CP}$ when the estimated concept presence exceeds a threshold.
> Concretely, we set
> $$\lambda_{CP} = 0.5 + 0.01*\max\\{CP_k(\cdot)-100,0\\}.$$
>
> Compared to uniformly increasing the guidance weight $\lambda$ from $1.5$ to $5.25$ in TraSCE, this adaptive strategy enables more effective concept removal while maintaining higher image fidelity.
>
> **Table 3. Results of TraSCE variants on the nudity removal task.**
> | Method | P4D ASR ↓ | Unlearn Atk ASR ↓ | MMA-Diff ASR ↓ | COCO CLIP ↑ | COCO FID ↓ |
> |---|---:|---:|---:|---:|---:|
> | TraSCE ($\lambda$=1.5) | 0.291 | 0.134 | 0.490 | **30.67** | **57.11** |
> | TraSCE ($\lambda$=5.25) | 0.225 | **0.113** | 0.506 | 30.18 | 61.70 |
> | TraSCE + CP | **0.199** | 0.120 | **0.388** | 30.43 | 57.20 |
>
> That said, the improvement remains limited, because this mechanism adjusts the guidance strength based only on CP, without accounting for how the guidance influences the subsequent denoising trajectory.

---

> > ### Author Rebuttal · Reviewer_qxe7 · 2026-04-04
> >
> > Thank the authors for providing the experiments to show the extension of this work.
> >
> > After carefully thinking, I would like to maintain my score.
> >
> > > The approach still depends on manually crafted negative prompts, which can be nontrivial to design in practice.
> >
> > I appreciate the authors' clever use of the guidance technique to address the safety scenario. However, I agree with RUtA's concerns about the design of user-specified negative prompts, which is still a difficult but critical issue.

---

> > > ### Author Response · Authors · 2026-04-08
> > >
> > > Thank you for your careful and constructive feedback. As we responded to Reviewer RUtA, we provide consistent results on automatically generated negative prompts.
> > >
> > > ## Regarding Negative Prompts
> > >
> > > In the original version of our paper, we selected negative prompts from prior literature. To address this concern more directly, we added experiments using negative prompts automatically generated by LLM.
> > > Specifically, we asked Grok to generate eight keywords most closely related to nudity and sexuality, and directly used them as negative prompts. Even with purely LLM-generated negative prompts, CRG remains effective and continues to perform strongly relative to the compared baselines.
> > >
> > > | Negative Prompts | Algorithm | P4D ↓ | Unlearn ↓ | MMA-Diff ↓ | CLIP ↑ | FID ↓ |
> > > |---|---|---:|---:|---:|---:|---:|
> > > | - | SD v1.4 | 0.854 | 0.655 | 0.949 | 31.10 | - |
> > > | Handcrafted | SAFREE | 0.529 | 0.275 | 0.611 | **31.11** | 55.79 |
> > > | Handcrafted | TraSCE | 0.291 | *0.134* | 0.490 | 30.67 | 57.11 |
> > > | Handcrafted | NP | 0.338 | 0.183 | 0.453 | 30.49 | *51.74* |
> > > | Handcrafted | DNG | *0.225* | 0.183 | **0.139** | 30.79 | 55.97 |
> > > | Handcrafted | CRG (OURS) | **0.026** | **0.059** | *0.164* | *30.85* | **36.05** |
> > > | GROK | SAFREE | 0.384 | 0.190 | 0.596 | *30.81* | 56.09 |
> > > | GROK | TraSCE | 0.265 | *0.120* | 0.508 | 30.60 | 57.91 |
> > > | GROK | NP | 0.331 | 0.176 | 0.423 | 30.35 | *51.79* |
> > > | GROK | DNG | *0.245* | 0.218 | **0.229** | 30.76 | 56.36 |
> > > | GROK | CRG (OURS) | **0.053** | **0.049** | *0.271* | **30.87** | **40.17** |
> > >
> > > [Grok 8 keywords] Nudity, Naked, Erotic, Sensual, Explicit, Sexual, Lewd, NSFW

---

### Official Review · Reviewer_cgQt · 2026-03-13

**Soundness:** 3
**Presentation:** 3
**Significance:** 3
**Originality:** 3
**Overall Recommendation:** 4
**Confidence:** 4

**Summary:**

The paper introduces Concept Removal Guidance (CRG), which is. a training-free plug-and-play method to remove unwanred-concept during diffusion steps. Comparing against many red-teaming benchmarks, the proposed method achieves state-of-the-art.

**Compliance With Llm Reviewing Policy:**

Affirmed.

**Ethical Review Concerns:**

The authors of this paper must have seen the nudity of some images in the evaluation process. I want to know whether (1) they have considered EU AI Act or is this legal? (2) Does the author try to obtain the ethical authorization? (3) whether all the authors are above 18?

**Ethics Expertise Needed:**

["Inappropriate Potential Applications & Impact (e.g., human rights concerns)", "Privacy and Security (e.g., personally identifiable information)"]

**Key Questions For Authors:**

Please see the weakness.

**Limitations:**

Yes

**Strengths And Weaknesses:**

Pros:
1. The paper has some insight. The analysis of Dynamic Negative Guidance is interesting and novel to me.
2. The method is novel and elegant.
3. The method is training-free and plug-and-play. Therefore, it is expected to be adopted by many diffusion models.
4. The appendix is exhaustive.

Overall. the paper is goodl

Cons:
1. Nudity is only one of the concerns. Whether the proposed method can be use to other concerns like in https://www.tiktok.com/community-guidelines/en.
2. The paper writing is bad. For instance, from Eq 6 to Eq 9 is hard to follow.Also, the abstract is hard to follow since there are some long sentences.
3. Have you ever considered the robustness?

---

> ### Author Rebuttal · Authors · 2026-03-31
>
> ## Additional Concept Removal
> We additionally evaluate CRG on **violence removal** using Ring-A-Bell with SD v1.4, 50-step DDPM sampling, and a shared negative prompt (“Bleeding, Blood, Gun, Weapon, Wounded”). Violent content is detected by the Q16 detector [1], and attack success rate (ASR) is reported in **Table 1** of [[link]](https://anonymous.4open.science/r/Rebuttal-CRG-B0EF/README.md).
> We also evaluate CRG on **identity removal** using SDXL with 50-step DDPM sampling. Evaluation was conducted using GPT-4o-mini and Qwen3-VL-235B-Instruct. We report ($Acc_e$) to measure target-identity removal, and ($Acc_u$) to assess preservation of non-target identities. As shown in **Table 2** of [[link]](https://anonymous.4open.science/r/Rebuttal-CRG-B0EF/README.md), the evaluation covers five celebrities, with Brad Pitt and Leonardo DiCaprio as removal targets.
>
> [1] Can Machines Help Us Answering Question 16 in Datasheets, and In Turn Reflecting on Inappropriate Content?
>
> ## Eq (6) to Eq (8)
> **Regarding Eq (6)**
> Our *concept presence* (CP) can be decomposed as the difference between two KL terms:
> $$CP(c-|c)\approx{\mathbb{E}\_{p_\theta(x_0|c)}\left[\log\frac{p_\theta(x_0|c)}{p_\theta(x_0)}\right]}
> -{\mathbb{E}\_{p_\theta(x_0|c)}\left[\log\frac{p_\theta(x_0|c)}{p_\theta(x_0|c-)}\right]}.$$
> - **The first term** is measured as the discrepancy between the *prompt-conditioned* distribution $p_\theta(x_0|c)$ and the *unconditional* distribution $p_\theta(x_0)$ through KL divergence.
> - **The second term** measures the discrepancy between the *prompt-conditioned* distribution $p_\theta(x_0|c)$ and the *negative concept* conditioned distribution $p_\theta(x_0)$ through KL divergence.
> Intuitively, $CP(c-|c)$ is *high* when $p_\theta(x_0|c)$ lies much closer to $p_\theta(x_0|c-)$ than to $p_\theta(x_0)$.
>
>
> **Regarding Eq (7)**
> Assuming a perfectly trained diffusion model, the model-induced trajectory distributions coincide with the true forward trajectory distributions in both the conditional and unconditional cases:
> $$p_\theta(x_{0:T}|c)=q(x_{0:T}|c),\quad p_\theta(x_{0:T})=q(x_{0:T}).$$
> Marginalizing over the intermediate variables $x_{1:T}$ then gives
> $$p_\theta(x_0|c)=q(x_0|c),\quad p_\theta(x_0)=q(x_0),$$
> and therefore the first MI term of Eq. (6) can be rewritten as
> $$D_\mathrm{KL}\big(p_\theta(x_0|c)\\|p_\theta(x_0)\big)=D_\mathrm{KL}\big(q(x_0|c)\\|q(x_0)\big).$$
> Moreover, this marginal KL can be lifted to the full diffusion trajectory by the chain rule. Since the forward noising process depends only on $x_0$ and is independent of $c$, we have
> $$q(x_{0:T}|c)=q(x_0|c)q(x_{1:T}|x_0),\quad q(x_{0:T})=q(x_0)q(x_{1:T}|x_0).$$
> Hence,
> $$D_\mathrm{KL}\big(q(x_{0:T}|c)\\|q(x_{0:T})\big)=\mathbb{E}\_{q(x_{0:T}|c)}\left[\log\frac{q(x_0|c)q(x_{1:T}|x_0)}{q(x_0)q(x_{1:T}|x_0)}\right]=D_\mathrm{KL}\big(q(x_0|c)\\|q(x_0)\big),$$
> where the common forward process term $q(x_{1:T}|x_0)$ cancels out. Combining this with the equalities above yields
> $$D_\mathrm{KL}\big(p_\theta(x_0|c)\\|p_\theta(x_0)\big)=D_\mathrm{KL}\big(p_\theta(x_{0:T}|c)\\|p_\theta(x_{0:T})\big).$$
> Using the standard reverse process factorization, the trajectory-level KL decomposes into a sum of per-step KL terms:
> $$D_\mathrm{KL}\big(p_\theta(x_{0:T}|c)\\|p_\theta(x_{0:T})\big)=\sum_{t=1}^{T}\mathbb{E}\_{p_\theta(x_t|c)}\left[ D_\mathrm{KL}\big(p_\theta(x_{t-1}|x_t,c)\\|p_\theta(x_{t-1}|x_t) \big)\right].$$
> Since both reverse transitions are Gaussian with the same covariance $\beta_t \mathbf{I}$, each per-step KL reduces to a scaled squared $\ell_2$ distance between the corresponding reverse means. Substituting the standard DDPM parameterization of the reverse mean in terms of $\epsilon_\theta$, we obtain
> $$D_\mathrm{KL}\big(p_\theta(x_0|c)\\|p_\theta(x_0)\big)=\mathbb{E}_{t,x,\epsilon}\left[\kappa_t\left\\|\epsilon_t^\theta(x_t,c)-\epsilon_t^\theta(x_t) \right\\|^2 \right],\quad\kappa_t=\frac{\beta_t T}{2\alpha_t(1-\bar{\alpha}_t)}.$$
>
>
> **Regarding Eq (8)**
> The second KL term in Eq. (6) admits the same derivation, with the *unconditional* distribution replaced by the *negative concept-conditioned* distribution $p_\theta(x_0|c-)$, and correspondingly $p_\theta(x_{0:T}|c-)$ at the trajectory level.
> We will revise the paper to reflect these comments and improve the overall readability.
>
> ## Robustness
> We further validate CRG across diverse concept removal settings, including violence, identity, and multi-concept removal. Additional results on multi-concept removal and nudity removal on PartiPrompts are provided in our responses to Reviewers “TE7Y” and “RUtA”.
>
> ## Ethical Concerns
> This study follows standard AI research ethics. All experiments use public models/datasets and synthetic images only, with no human participants or identifiable personal data; thus, ethics approval was not required. Our method supports model safety and constitutes a low-risk academic use aligned with the EU AI Act. All authors are above 23.

---

### Decision · Program_Chairs · 2026-04-30

**Decision:**

Accept (spotlight)

**Comment:**

This paper received Accept x2, WA x1 (the reviewer missed rebuttal acknowledgement), and WR x1. The reviewer who initially gave WR acknowledged their concerns are fully resolved and did not provide substantive counter-arguments to the rebuttal.  After carefully reviewing the rebuttals and reviewer comments, the AC recommends **accept**.

This paper makes a strong contribution by addressing an important AI safety problem (ie, content removal) through a highly practical design (training-free and plug-and-play design), supported by solid empirical results and math grounding.  The authors also provided a comprehensive rebuttal, with additional results from a new human eval study that effectively addresses the reviewers' primary concerns.

For the final version, the authors are encouraged to incorporate the reviewers' feedback to further improve paper clarity and empirical rigor, particularly strengthening the presentation of the human eval study and the robustness experiments, and discussing potential extensions to multi-concept removal.